# Determinants of PrEP Uptake, Intention and Awareness in the Netherlands: A Socio-Spatial Analysis

**DOI:** 10.3390/ijerph19148829

**Published:** 2022-07-20

**Authors:** Haoyi Wang, Oladipupo Shobowale, Chantal den Daas, Eline Op de Coul, Bouko Bakker, Aryanti Radyowijati, Koenraad Vermey, Arjan van Bijnen, Wim Zuilhof, Kai J. Jonas

**Affiliations:** 1Department of Work and Social Psychology, Maastricht University, 6200 ER Maastricht, The Netherlands; haoyi.wang@maastrichtuniversity.nl (H.W.); o.shobowale@maastrichtuniversity.nl (O.S.); 2Health Psychology, University of Aberdeen, Aberdeen AB24 3 FX, UK; chantal.dendaas@abdn.ac.uk; 3Centre for Infectious Disease Control, National Institute for Public Health and the Environment (RIVM), 3721 MA Bilthoven, The Netherlands; eline.op.de.coul@rivm.nl; 4Rutgers, 3511 MJ Utrecht, The Netherlands; b.bakker@rutgers.nl; 5ResultsinHealth, 2352 BN Leiderdorp, The Netherlands; aryanti@resultsinhealth.org; 6Soa Aids Netherlands, 1014 AX Amsterdam, The Netherlands; kvermey@soaaids.nl (K.V.); avanbijnen@soaaids.nl (A.v.B.); wzuilhof@soaaids.nl (W.Z.)

**Keywords:** MSM, PrEP use/uptake, socio-spatial analysis, HIV prevention

## Abstract

PrEP uptake in the Netherlands is growing but remains at suboptimal levels. Hence, the analysis of hurdles is paramount. Given the initial focus of PrEP provision among men-who-have-sex-with-men (MSM) via a demonstration project that was launched in June 2015, AmPrEP in Amsterdam, and pharmacies in the main urban areas (so called “Randstad”, entailing Amsterdam, Utrecht, Leiden, The Hague and Rotterdam), investigating regional differences is necessary. This study seeks to unravel regional differences jointly with the psycho-social determinants of PrEP uptake. This cross-sectional study included 3232 HIV-negative MSM recruited via the Dutch subsample of the European-MSM-Internet-Survey in late 2017 (EMIS-2017), which aimed to inform interventions for MSM who are highly affected by infections with HIV and other sexually transmitted infections. Prevalence and the standardised prevalence ratio (SPR) of PrEP awareness, intention and uptake were measured on a regional level (Randstad vs. the rest of the country). Multi-level logistic modelling was conducted to identify the association of PrEP uptake with PrEP awareness and intention, socio-demographic, psycho-social determinants and random effects from regional differences. MSM from the Randstad used more PrEP (SPR = 1.4 vs. 0.7) compared to the rest of the country, but there were minor differences for awareness and intention. The regional distinction was estimated to explain 4.6% of the PrEP use variance. We observed a greater influence from PrEP intention (aOR = 4.5, 95% CI 2.0–10.1), while there was limited influence from the awareness of PrEP (aOR = 0.4, 95% CI 0.04–4.4). Lower education (aOR = 0.4, 95% CI 0.2–0.9) was negatively associated with PrEP uptake; however, no significant difference was found between middle (aOR = 1.2, 95% CI 0.7–2.0) and high education. We showed that regional differences—MSM in non-urban regions—and other psycho-social determinants account for lower PrEP uptake. Based on these findings, more fine-tuned PrEP access with a focus on non-urban regions can be implemented, and tailored campaigns increasing intention/use can be conducted among target populations.

## 1. Introduction

While new HIV infections are declining overall in the Netherlands, HIV transmission remains a major health threat among men-who-have-sex-with-men (MSM) [1]. As a biomedical intervention and part of comprehensive prevention measures, pre-exposure prophylaxis (PrEP) is highly effective in preventing HIV-negative MSM from acquiring HIV [2,3,4,5,6]. In response, PrEP was formally introduced in Europe in 2016 and was subsequently included in the Dutch national delivery plan in September 2019 through the sexual health clinics of the Public Health Services (GGD) [1,7,8,9]. It is estimated that around 10,000 MSM meet the eligibility criteria for PrEP in the Netherlands [4]. Yet, PrEP uptake remains suboptimal in the Netherlands [10], which underlines the need for better tailored PrEP provision. To further promote PrEP uptake, an improved quality of the epidemiologic surveillance of the current PrEP uptake and intention among MSM could better inform the national and local PrEP programming, including allocation and access. This paper seeks to provide such evidence by means of suggesting a socio-spatial analysis investigating regional differences in conjunction with individual-level determinants of PrEP uptake among MSM, in the sample context of the Netherlands.

### Estimating PrEP Awareness, Intention and PrEP Uptake

In most countries, including the Netherlands, there are no centralised and comprehensive data available that allow the determination of actual PrEP use. Hence, PrEP use data are often based on self-report only, or actual use data are split up across a number of registries and stored at different data aggregation levels which makes it impossible to determine subpopulations or regions that require special attention to close the PrEP uptake gap.

To stay within the Dutch example, delivering PrEP remains challenging [11], especially when PrEP care is offered by decentralised healthcare providers (the 25 GGDs in the Netherlands), by GPs and by private clinics across the country. The 8500 slots for PrEP users to be supplied by the GGDs are considered to be more or less taken, but there is little information available about PrEP users accessing PrEP via their GPs or private clinics [12]. To improve PrEP access, it is important to understand the needs of MSM such as the awareness and intention to use PrEP, preferably specified by geographical regions. Consequently, PrEP access and other related resources can be allocated appropriately. Thus, PrEP self-reported uptake data can be complemented by awareness and intention data, too.

To date, PrEP intention among MSM data have been mostly gathered among MSM residing in Amsterdam [7,13,14,15]. Cross-sectional data from these Amsterdam-based studies revealed that PrEP intention among MSM increased from 13% (high intention to use) during the initial assessment in 2013 [13] to a higher proportion of definite intention of almost 40% in 2016 [7]. To corroborate these findings, a significant increasing trend of informal PrEP uptake among Dutch MSM residing in Amsterdam from 2015 to 2017 was also observed [16,17]. While these temporal trends can provide a good picture of PrEP use among MSM in Amsterdam over time, they are also a glaring reminder of the missing data and information from other cities and regions in the Netherlands.

Hence, geo-spatial data on differences in PrEP awareness, intention and uptake across the Netherlands have the potential to complement the epidemiologic trend of PrEP uptake by providing a more comprehensive perspective. The spatial distribution of current PrEP awareness, intention and uptake can be useful to evaluate the national programme and can help to tailor local PrEP access in order to prioritise where PrEP implementation should be further strengthened [11,18]. In the Netherlands, the characterisation of the residence of an individual is usually divided into the main urban area (so called “Randstad”, entailing Amsterdam, Utrecht, Leiden, The Hague and Rotterdam, more details in Section 2) and the rest of the country (ROC) [7,14,19]. However, a geo-spatial distribution of the Randstad-ROC level of PrEP awareness, intention and uptake among MSM has yet to be analysed in the Netherlands, mainly due to the limited availability of PrEP prior to its structural introduction [1]. However, available PrEP uptake data from national surveys, which aim to access and improve health among MSM, such as the European MSM Internet Survey 2017 (EMIS-2017) can be employed to amend the current data and to fill knowledge gaps. Therefore, we sought to investigate how PrEP awareness, intention and uptake among MSM are distributed under this geo-spatial framework in the Netherlands.

Furthermore, it is important to jointly understand the psycho-social facilitators and hurdles of PrEP uptake together with spatial trends to unravel co-variation. So far, in the Global North, studies have mostly examined the individual-level determinants of PrEP uptake such as HIV transmission risk; prior STI diagnoses; sexualized substance use (chemsex); and socio-demographic factors, such as age, race/ethnicity, education, employment status and income [7,18,20,21,22]. In addition, urban/rural differences in PrEP awareness, intention and uptake have been investigated both quantitatively [23,24] and qualitatively [25], showing that MSM living in a more urbanised region have a higher chance of being aware, showing intention and using PrEP. However, to our knowledge, no attention has been given to investigating these determinants jointly with the spatial distribution of PrEP uptake among MSM, even though both spatial and socio-ecological determinants may influence PrEP uptake trends. We thus sought to explore whether these individual determinants of PrEP uptake and the spatial variation of PrEP uptake can provide a deeper understanding of PrEP uptake when combined.

To gain insights on the spatial trend of PrEP awareness, intention and uptake in the Netherlands, we aimed to conduct a socio-spatial analysis using data collected by the EMIS-2017. Specifically, we sought to investigate regional differences in conjunction with individual-level determinants of PrEP uptake under a socio-spatial framework.

## 2. Materials and Methods

### 2.1. Study Population and Study Design

This study has a cross-sectional design. All men in this study were originally recruited between 19 October 2017 and 30 January 2018 via the European MSM Internet Survey (EMIS-2017, www.emis2017.eu, accessed on 17 May 2022). The EMIS-2017 was an anonymous, self-administered and cross-sectional online survey conducted across Europe to inform interventions for MSM who are highly affected by infections with HIV and other sexually transmitted infections (STIs) [26]. Ethical approval for this survey was obtained from the Observational Research Ethics Committee at the London School of Hygiene & Tropical Medicine (review reference 14421/RR/8805). We used data from the Netherlands-based respondents from the EMIS-2017 (participation eligibility and further details about the EMIS-2017 survey can be found in the published EMIS-2017′s design and methods report [26]). In our analysis, we only included data from respondents who indicated they were living in the Netherlands and were living without HIV at the time of data collection.

### 2.2. Study Settings and Measures

All measures were self-reported. All men were assigned to geo-regions in the Netherlands, in this study the Randstad and the ROC, based on the area of residence as identified by the postal code provided within the survey. The Randstad entails the agglomeration of cities in the west of the Netherlands, in particular Amsterdam, Utrecht, Leiden, The Hague and Rotterdam. As the economic and political centre of the Netherlands, the Randstad accounts for approximately 50% of the national population and the majority of the newly-diagnosed HIV infections [1]. The remainder of postcode regions were coded together and referred to as the Rest of the Country (ROC).

### 2.3. Outcomes and Covariates

In the descriptive analysis, the outcome variables were PrEP awareness, PrEP intention and PrEP uptake. We dichotomised these variables into “yes”/“no”. To take the potential influence from socio-demographic determinants into account, age in 10-year-bands (eight groups in total, see Appendix A Appendix A); education level (categorised as low, median and high); employment status ((self-)employed, unemployed, student and retired); and income status (low, median and high) were used as covariates to build the standardisation matrix (for more details see Section 2.4.1).

In the multi-level modelling analysis, PrEP uptake (yes/no) was the only outcome variable. Covariates were PrEP awareness (yes/no); PrEP intention (yes/no); age in years; education level; employment status; income status; sexual orientation (homosexual/bisexual and straight or other); disclosure of sexual orientation (non-disclosed, low-partially disclosed, high-partially disclosed and fully); recency of having anal sex with another man; safe-sex efficacy (decisions to have safe sex and decisions to reject sex); PrEP knowledge; HIV knowledge; depression/anxiety level (PHQ-4) [27]; alcohol dependency (CAGE-4) [28]; condomless anal intercourse (CAI) with non-steady partner (yes/no); CAI with steady partner (yes/no); CAI with non-steady partner living with HIV (yes/no); STI status (ever/never); transactional sex recency (never, ever but more than 12 months ago, and ever within the previous 12 months); injecting drug use (IDU, yes/no); and chemsex recency. See Appendix A Appendix A for more details.

### 2.4. Data Analysis

#### 2.4.1. Descriptive Analysis

We first estimated and compared the crude prevalence and standardised prevalence ratio (SPR) of PrEP awareness, intention and uptake between the Randstad and ROC. We used the Wilson score interval method to estimate the 95% confidence intervals (95% CI). The SPR of PrEP awareness, intention and uptake in this study were standardised by the socio-demographic variables’ matrix (participant’s 10-year age group (eight groups), employment status (four groups), income status (three groups) and education level (three groups), Appendix A Appendix A) in a total of 288 (=8 × 4 × 3 × 3) strata with an indirect standardisation approach:(1)$${\mathrm{SPR}}_{i}=Y_{i}/E_{i}$$
(2)$$E_{i}=r^{\left(s\right)}n^{\left(i\right)},$$
where $r^{\left(s\right)}$ is the overall prevalence in all regions, and $n^{\left(i\right)}$ presents the study sample size of region i. This allows us to compare the risk levels in different regions if region i has higher (SPR > 1), equal (SPR = 1) or lower (SPR < 1) risk than the overall prevalence in the total study population.

To account for the Dutch PrEP eligibility criteria, we conducted a sensitivity analysis which only included men who met the Dutch PrEP eligibility criteria: (1) men who were not always using condoms during anal intercourse with both steady/non-steady partners; (2) men who have a relationship with a partner living with HIV; (3) men who had been diagnosed with any type of STIs (syphilis, chlamydia or gonorrhoea) or had ever had transactional sex or ever reported injection drug use.

#### 2.4.2. Multi-Level Analysis

Under the assumption that there would be an influence of region of residence (Randstad and ROC) on PrEP uptake, a generalised logistic mixed modelling analysis was conducted to investigate both the variation between regions and the socio-demographic, behavioural and psycho-social determinants of this variation [29]. Accordingly, we considered two levels—the Dutch HIV-negative MSM included in this study at level 1 nested within the two geo-regions at level 2, for a generalised logistic mixed model. In the multi-level analysis, we included those men who did not meet the eligibility criteria in the multi-level analysis, as the eligibility criteria were reflected by the included covariates.

In the multi-level models, we first fitted a null two-level model with only a random intercept to ascertain the variance between geo-regions. Next, we added all explanatory variables included in this study to the two-level model with only a random intercept to identify determinants. A manual stepwise backward selection approach was applied to select the explanatory variables. We retained all variables with *p* < 0.05. We also retained PrEP awareness and PrEP intention in the final model regardless of the statistical significance since they were the key psycho-social variable of interest. The Akaike information criterion (AIC) was used to access the goodness of fit of the models. Associations were represented by adjusted odds ratios (aOR) with 95% CIs. We used marginal R^2^ to access the variance of the fixed effects. Additionally, conditional R^2^ was applied to estimate the variance taking both the fixed and the random effects into account. Interclass-correlation (ICC) was calculated for the final model to estimate the influence from the geo-regions. All analyses were conducted with R software (version R 4.0.4 GUI 1.70) (R Foundation for Statistical Computing, Vienna, Austria).

## 3. Results

### 3.1. Study Population Characteristics

The EMIS recruited 3851 MSM in the Netherlands. We excluded 619 individuals living with HIV (PLHIV, 16.1%), and included 3232 men who were HIV negative in our study (for the sensitivity analysis accounting for the Dutch PrEP eligibility criteria, 2424 men were included). Among these men, 35.6% (1150 of 3232) were living in the Randstad region in the Netherlands, and 64.4% (2082 of 3232) were residents from the ROC. The median age of these respondents was 43 (range 16–87). Overall, 63.2% (2044 of 3232) of the respondents had at least an HBO degree (equivalent to college education), while only 12.9% (418 of 3232) of these men did not complete high school. In terms of financial situation, 67.6% (2184 of 3232) of the respondents regarded themselves as having a comfortable/very comfortable life based on their income, compared to only 8.3% (269 of 3232) who were struggling to make do. The majority (74.4%) was employed, and 10% of the participants were students and 6% were retired. Information on other participant characteristics by geo-regions can be found in Table 1. Detailed age-stratified distributions of the education level, income status and employment status can be found in Appendix A Appendix A.

### 3.2. Prevalence and Standardised Prevalence Ratio of PrEP Awareness, Intention and Uptake

To estimate the crude prevalence and SPR of PrEP awareness, intention and uptake among the respondents, Table 2 describes the distributions in the Randstad and the ROC in the Netherlands. Almost eight percent of the MSM in the Randstad region used PrEP (7.9%, 95% CI 6.5–9.6), which was more than twice as high as in the rest of the Netherlands (3.1%, 95% CI 2.4–3.9). After standardising by 10-year age bands, education level, income status and employment status, the SPR of MSM in the Randstad was estimated to be much higher than the ROC (SPR 1.4 vs. 0.7).

More than 90% of the MSM living in the Randstad were aware of PrEP (90.1%, 95% CI 89.0−92.4%), compared to 78% in the ROC (78.1%, 95% CI 76.2−79.8%). However, after standardising, the SPR of PrEP awareness among MSM living in the Randstad was similar to the ROC (SPR 1.1 vs. 0.9). Similar proportions of MSM intended to use PrEP in both regions (Randstad: 46.3%, 95% CI 43.4−49.2%, SPR = 1.02; ROC: 45.4%, 95% CI 43.3−47.6%, SPR = 0.94).

In the sensitivity analysis, after excluding men who did not meet the Dutch PrEP eligibility criteria, we observed a same overall trend but slightly higher prevalence and SPR of PrEP awareness, intention and uptake in the Randstad (see Appendix A Appendix A for full details).

### 3.3. Multi-Level Analysis

#### 3.3.1. Between Region Variances

To investigate the impact of place of residence, we conducted multi-level analyses. Information and comparisons between the full model and the final model can be found in Appendix A Appendix A. The ICC of the level 2 random effect from the final model, compared to the full model, decreased to 0.046 (τ_00_ = 0.15, σ^2^ = 3.29) from 0.054, which indicates that 4.6% of the residual variation in the PrEP uptake among HIV-negative MSM in the Netherlands can be attributed to the unobserved regional characteristics between the Randstad and the ROC (see Appendix A Appendix A). This depicts a small but non-ignorable variance between the Randstad region and the rest of the country. A detailed performance check of the final model, including collinearity between variables, can be found in Appendix A Appendix A.

#### 3.3.2. Random Intercept Model

To investigate the determinants of PrEP uptake, Table 3 presents the association between explanatory variables and PrEP uptake by the final model. According to the final model, the greater influence from PrEP intention (aOR = 4.5, 95% CI 2.0–10.1) was estimated compared to the limited influence from the awareness of PrEP (aOR = 0.4, 95% CI 0.04–4.4).

Among behavioural determinants, having CAI with non-steady partners (aOR = 2.3, 95% CI 1.1–4.8), having CAI with HIV positive partners (aOR = 1.89, 95% CI 1.2 to 3.0), having ever been diagnosed with an STI (OR = 2.1, 95% CI 1.1 to 3.9) and injecting drugs (aOR = 2.4, 95% CI 1.2–4.5) were estimated to have positive associations with PrEP uptake. Compared to those who bought/sold sex within the last 12 months, men who had transactional sex more than 12 months ago showed a lower aOR of 0.47 (95% CI 0.25–0.90), while no significant difference was detected between men who had recent transactional sex and those who never have transactional sex (aOR = 0, 95% CI 0-Inf).

Psycho-social determinants including the decision to have safe sex (aOR = 1.3, 95% CI 1.0–1.6), and having sufficient PrEP knowledge (aOR = 7.0, 95% CI 4.1–12.0) were found to be significantly associated with higher odds of PrEP uptake. However, the decision to reject sex (aOR = 0.76, 95% CI 0.56–1.03), HIV knowledge (aOR = 1.11, 95% CI 0.60–2.04), PHQ-4 (depression/anxiety level, aOR = 1.09, 95% CI 0.79–1.51) and CAGE-4 (alcohol dependency, aOR = 1.33, 95% CI 0.67–2.63) were not significantly associated with PrEP uptake.

It is worth mentioning that, among socio-demographic determinants, only low education (aOR = 0.4, 95% CI 0.2–0.9) was found to be negatively associated with PrEP use. No significant difference was found between median and high education (aOR = 1.2, 95% CI 0.7–2.0). Age, employment status and perceived financial status were found to be not relevant to explain the PrEP uptake in our sample (Full Model, Appendix A Appendix A).

## 4. Discussion

Using data collected from MSM in the EMIS-2017 survey, we explored and analysed the spatial distribution of PrEP awareness, intention and uptake in the Netherlands, based on a socio-spatial structure of the main urban area (Randstad) vs. the rest of the country (ROC). Our analysis revealed heterogeneity in the spatial distribution of PrEP uptake in the Netherlands.

In our estimations of prevalence and SPR of PrEP awareness, intention and uptake, we observed a high prevalence of the awareness and intention of PrEP but a much lower uptake of PrEP among MSM (around 8% in the Randstad and 3% in the ROC) in the Netherlands. Our results confirmed that there was a gap to bridge to reach the cost-effectiveness requirements of 10% as set by Nichols et al. [30], even though our estimation in the sensitivity analysis indicated a PrEP uptake of more than 10% among those men who met the Dutch PrEP eligibility criteria in the Randstad (10.1%, see Appendix A Appendix A).

Despite the proportion of awareness and intention of PrEP being similar between the Randstad region and the ROC in the Netherlands, the prevalence of the PrEP uptake between the Randstad region and the ROC differed by almost two times. In the Randstad vs. ROC-level analysis, our generalised logistic models confirmed the observation that a small but non-ignorable proportion of variance of PrEP uptake among MSM in the Netherlands can be explained by whether the residence place is in the Randstad area or not. One possible reason may be the unequal geographic coverage of the PrEP provision due to a lack of access sites [11,31] in the ROC compared to the Randstad region. To explain this finding, it is noteworthy to mention that besides individual access via a GP (at much higher costs), PrEP was only available via the Amsterdam PrEP (AMPrEP) demonstration project or some activist groups [32] prior to its structural introduction via STI clinics [1,33]. Another possible reason to explain this difference may be the variance in believes/attitudes about PrEP among STI/HIV professionals [34] in the Randstad region and the ROC [19]. In addition, some psycho-social and socio-demographic determinants may also explain our findings. Determinants such as low education level, insufficient PrEP knowledge and injecting drug use were higher in the ROC compared to the Randstad (Table 1). These may suggest psycho-social/socio-demographic barriers for PrEP access in the ROC compared to the Randstad. We thus call for public health efforts and further interventions to be allocated to the ROC which focus on these barriers of PrEP access to better tailor localised HIV prevention.

We further explored the determinants of PrEP uptake in the Netherlands in the socio-spatial structure context. We found that compared to being aware of PrEP, the intention may facilitate the PrEP uptake in the Netherlands. In addition, similar to the determinants of PrEP interest in the Netherlands reported by van Dijk et al. we observed PrEP knowledge and CAI as determinants [7]. However, instead of engaging in chemsex [7], the ever-injecting drug use is significantly associated with PrEP uptake. In addition, both Coyer et al. and van Dijk et al. reported that PrEP users did not differ from non-PrEP users in terms of socio-demographic characteristics [14,16]. Our results confirmed their findings except for participants with low education, who had a lower chance of PrEP uptake compared to MSM who had a high education in our data. One possible reason may be the unequal economic environment, which shapes the distribution of resources and barriers within a society [35], between the Randstad and the ROC.

### Strengths and Limitations

One major strength of this study was the socio-spatial analysis of PrEP uptake, which combines both Randstad and the ROC into the inferential process for the first time. It not only offers national and local PrEP implementation teams with a spatial distribution, but also examines the determinants of PrEP uptake while considering spatial heterogeneity. Our methodology can thus be employed in the future of PrEP surveillance when comparing determinants between different geo-regions. We thus further expect our results will retain their relevance in the future and provide evidence for the future investigations of PrEP uptake in any context, also outside of the Netherlands. In addition, our sensitivity analysis on the with/without non-eligible men can be considered evidence for the role of the PrEP eligibility criteria, even though the differences between the analyses were minor in this sample context. Future studies thus should not ignore the influence of the PrEP eligibility when investigating the epidemiology of PrEP uptake.

There are a few limitations in our study. One major limitation may be that our data are not devoid of biases. For example, the limited sampling contexts, above and beyond the self-report nature of the survey, have been discussed in the previous EMIS methods paper [26]. Another limitation may be the lack of data for our discussion of the acceptability of PrEP prescription among STI/HIV specialists. Given the fact that STI/HIV specialists from the Randstad region have a higher acceptability of PrEP [19] before the nationwide structural PrEP introduction, we hypothesised a possible association between the regional level of PrEP support among STI/HIV specialists and the PrEP uptake among MSM in the Netherlands. However, we cannot conclude this association, again, due to lack of data. Therefore, future studies should collect information on PrEP beliefs and attitudes from the STI/HIV specialists and integrate this information in modelling analyses. In addition, we only investigated the determinants of PrEP uptake through our generalised logistic mixed modelling analysis. The lack of modelling analysis of the determinants of PrEP awareness and PrEP intention may be another limitation. Our findings, especially the lower prevalence of PrEP intention than PrEP awareness for both the Randstad region and the ROC, suggested more studies are needed to explore strategies to increase the intention of PrEP from being aware of PrEP among MSM. More efforts are thus needed from the healthcare professionals to bridge the needs and intention from the PrEP promotion and access in the Netherlands. One major methodological limitation in this study could be the lack of a concise spatial structure of geo-information of the Randstad-ROC structure in the Netherlands. A more robust methodology, such as a small area estimation analysis, thus cannot be conducted [36]. The available level of geo-information made a prior distribution assignment, or honing in on other areal information, impossible. Consequently, our spatial heterogeneity estimation for the Randstad vs. the ROC may thus be less robust. However, with the strength from the generalised logistic modelling, given the fact that we still succeeded in picking up the spatial variances, the message of unequal spatial distribution of PrEP uptake in the Netherlands remains valid. Hence, future studies should zoom in on a lower geographical level, such as the GGD regional level, to investigate a more concise spatial distribution through a more comprehensive model such as the Bayesian spatial modelling analysis. Another major methodological limitation may be the lack of data on the temporal dimension of PrEP awareness, intention and uptake. This may limit the scope of this study in that we cannot conclude how the PrEP uptake changed from a spatio-temporal perspective. We acknowledge that our data were collected prior to the structural introduction of PrEP in the Netherlands, but this factor is less relevant for the general applicability of the approach that we introduced here. Naturally, a replication with more recent data remains desirable. The spatial distribution of PrEP awareness, intention and uptake among MSM may thus change due to the effort of the PrEP promotion in the Netherlands through the GGDs [37]. Therefore, future studies should take the temporal variance, especially data prior to/post the structural introduction, together with the spatial distribution of PrEP awareness, intention and uptake into account to offer a more dynamic epidemiologic picture.

## 5. Conclusions

With the estimated differences between the Randstad region and the rest of the country in the Netherlands on PrEP uptake, together with the psycho-social determinants of PrEP uptake, results from this study can inform the current national PrEP program in the Netherlands to evaluate the PrEP implementation and allocation, and to identify regions and populations that require attention to close the PrEP uptake gap in the Netherlands. Resources and attention should be allocated more towards the rest of the country compared to the Randstad to bridge the geographic gaps of PrEP awareness, intention and uptake among MSM.

## Figures and Tables

**Table 1 ijerph-19-08829-t001:** Participant characteristics by geo-regions.

Variables	Randstad		ROC		Overall	
	N	%	N	%	N	%
Age *	43	16–80	43	16–87	43	16–87
Education level						
High	820	71.3	1224	58.8	2044	63.2
Median	220	19.1	550	26.4	770	23.8
Low	110	9.6	308	14.8	418	12.9
Employment status						
Employed	908	79.2	1495	72.0	240 3	74.6
Retired	62	5.4	133	6.4	195	6.1
Student	94	8.2	225	10.8	319	9.9
Unemployed	83	7.2	222	10.7	305	9.5
Financial coping						
(Really) comfortable	814	70.8	1370	65.8	2184	67.6
Neutral	238	20.7	541	26.0	269	8.3
(Really) struggling	98	8.5	171	8.3	779	24.1
Sexual identity						
Gay	1021	88.8	1685	80.9	2706	83.7
Bisexual	83	7.2	300	14.4	383	11.9
Straight or other	46	4.0	97	4.7	143	4.4
Disclosure of sexual orientation (outness)						
All or almost all	862	75.0	1362	65.4	2224	68.8
More than half	136	11.8	214	10.3	350	10.8
Few or less than half	145	12.6	485	23.3	630	19.5
None	7	0.6	21	1.0	28	0.9
Having anal sex with another man recency						
Within 12 months	958	83.3	1674	80.4	2632	81.4
Never or more than 12 months	192	16.7	408	19.6	600	18.6
Decision to have safe sex #	5	NA	4	NA	4	NA
Decision to reject sex #	4	NA	5	NA	4	NA
PrEP knowledge #	2	NA	0	NA	0	NA
HIV knowledge #	3	NA	3	NA	3	NA
PHQ−4 (Depression/anxiety level) #	1	NA	1	NA	1	NA
CAGE−4 (Alcohol dependency) #	1	NA	1	NA	1	NA
CAI with non-steady partner						
Yes	379	33.0	633	30.4	1012	31.3
No	771	67.0	1449	69.6	2220	68.7
CAI with steady partner						
Yes	143	12.4	272	13.1	415	12.8
No	1007	87.6	1810	86.9	2817	87.2
CAI with non-steady partner with diagnosed HIV						
Yes	129	28.2	164	19.8	293	22.8
No	329	71.8	665	80.2	994	77.2
Ever diagnosed with syphilis, gonorrhoea or chlamydia						
Ever	692	60.2	1046	50.2	1738	53.8
Never	458	39.8	1036	49.8	1494	46.2
Transactional sex recency						
Ever, within 12 months	126	11.0	225	10.8	351	10.9
Ever, more than 12 months	1012	88.0	1805	86.7	2817	87.1
Never	12	1.0	52	2.5	64	2.0
Injecting drug use						
Ever	59	5.1	110	5.3	169	5.2
Never	1091	94.9	1972	94.7	3063	94.8
Chemsex recency #	8	NA	8	NA	8	NA

Notes: Self-efficacy_1 = The sex I have is always as safe as I want it to be. Self-efficacy_2 = I find it easy to say “no” to sex I don’t want. *: value presented in median and range. #: values presented in mode. ROC = rest of the country. NA = not applicable. CAI = condomless anal intercourse.

**Table 2 ijerph-19-08829-t002:** Estimated prevalence and standardised prevalence ratio, Randstad vs. the rest of the country (ROC), 2017.

Region	PrEP Uptake		PrEP Awareness		PrEP Intention	
	Prevalence (%, 95% CI)	SPR *	Prevalence (%, 95% CI)	SPR *	Prevalence (%, 95% CI)	SPR *
Randstad	7.91 (6.49–9.62)	1.41	90.84 (89.02–92.38)	1.11	46.26 (43.4–49.15)	1.02
ROC	3.07 (2.41–3.91)	0.71	78.08 (76.23–79.83)	0.94	45.44 (43.31–47.58)	0.94

Note: Data are crude prevalence (95% confidence interval). Data are not age/education level/income status/employment status- standardised. * Data are age/education level/income status/employment status- standardised. ROC = rest of the country.

**Table 3 ijerph-19-08829-t003:** Generalised logistic mixed final models: variation between regions and the determinants of PrEP uptake.

Predictors	Final Model PrEP Uptake(AIC = 568.6)		
	aOR	95% CI	*p*-Value
PrEP awareness			
No	Ref.	-	-
Yes	0.4	0.04–4.06	0.437
PrEP intention			
No	Ref.	-	-
Yes	4.46	1.95–10.20	<0.001
Decision to have safe sex	1.29	1.03–1.61	0.028
PrEP knowledge	7.34	4.28–12.56	<0.001
CAI with non-steady partner			
No	Ref.	-	-
Yes	2.3	1.11–4.77	0.024
CAI with non-steady partner with diagnosed HIV			
No	Ref.	-	-
Yes	1.97	1.26–3.10	0.003
Ever diagnosed with syphilis, gonorrhoea or chlamydia			
Never	Ref.	-	-
Ever	2.14	1.15–3.98	0.016
Transactional sex			
Ever within the previous 12 months	Ref.	-	-
Ever but longer than 12 months	0.52	0.29–0.96	0.036
Never	0	0.00–Inf	0.993
Injecting drug use			
Never	Ref.	-	-
Ever	2.28	1.20–4.34	0.012
Education			
High	Ref.	-	-
Median	1.24	0.76–2.03	0.392
Low	0.4	0.17–0.92	0.03
Random effects			
σ^2^	3.29		
τ_00_	0.15_randstad_		
ICC	0.046		
Marginal R^2^/Conditional R^2^	0.706/0.719		

Note: AIC = Akaike information criterion. aOR = adjusted odds ratio. 95% CI = 95% confident intervals. CAI = condomless anal intercourse. Information from the full model can be found in Appendix A Appendix A.

## Data Availability

Data are available upon request.

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
