# Peer review of "Determinants of PrEP Uptake, Intention and Awareness in the Netherlands: A Socio-Spatial Analysis"

_ijerph, 2022, doi:10.3390/ijerph19148829_

Round 1

Reviewer 1 Report

The reviewed ms reports a multilevel modeling study that aims to identify determinants of PrEP uptake in the Netherlands, with specific aims to unreveal regional differences and also compares several psycho-social determinants. The authors report interesting finding for PrEP uptake determinants that could facilitate and tailore PrEP programs  among target populations. The ms is well-writen and data are clearly presented and could be of interest for IJERPH readers. 

Reviewer 2 Report

The study is well justified and timely to context, given the recent introduction of PrEP in the Netherlands.  The aim to perform a socio-spatial analysis is admirable.  The findings are not surprising reporting marginal socio-spatial differences  between urbanized and rest of the country regarding uptake, and in favor of the urban areas.  My enthusiasm for the  study is dampened by that the “spatial” analysis differentiation is  pretty gross (urban vs rest of country), which takes away from the study’s explanatory power.  As a matter of fact, most of the analysis do not capture the scope and prescription possible with geospatial analysis.  The analysis are much stronger at the individual  rather than spatial difference levels.  The observed “small” difference speaks to the fact that the study design is underpowered to capture spatial effects that would explain any differences.  The study’s merit is providing preliminary evidence, and hopefully to be followed up with more rigorous geospatial study.    

Reviewer 3 Report

Major comments:

1. The introductory paragraph jumps all over - it is all good information, but it is unclear/inappropriate what is being compared to lay the foundation for this analysis. For example, you say PrEP is estimated to be cost-effective in the Netherlands over a 30-year time frame (citing a paper from 2016) and then say "In response, PrEP has been formally introduced in Europe in 2016." I can't imagine PrEP was introduced in Europe in response to this Dutch paper published in 2016. Similarly, you say, "It is estimated that around 10,000 MSM meet the eligibility criteria for PrEP in the Netherlands," and immediately follow that with "Yet, PrEP uptake remains suboptimal in many countries." That is true, but seems irrelevant. Is it sub-optimal in the Netherlands? That's what you should be explaining and citing. In the second paragraph, you describe France as an example. I highly suggest revising this paper to stay coherently within the Dutch context, except where necessary to expand to Europe to support your argument.

2. Your third paragraph says that "delivering PrEP remains challenging," especially with a decentralized health system - "thus, PrEP self-reported uptake data has to be complemented by awareness and intention data, too." If PrEP delivery is challenging in the Netherlands, why is it important to know about patient awareness and intention to take PrEP? This point actually seems to undermine your argument.

3. While you (presumably accurately) make the point that studies examining individual-level determinants of PrEP uptake rarely investigate those determinants jointly with the spatial distribution of PrEP uptake among MSM, there are certainly numerous studies in the literature that examine regional and urban/rural differences in PrEP uptake (e.g. Rossiter 2021; Sarno 2021; and Walters 2022; all in AIDS Behav, just in a quick search). Are those applicable to the Dutch context? Do you have reason to believe it is different in the Netherlands that in these other places? How does this joint analysis contribute something that cannot already be understood by existing research? This is what I would expect to see in the Introduction of this paper.

4. You say that backward selection using AIC was used to finalize your two-level logistic model, but do not say what variables (from a VERY long list) were included in the final model in the Methods section. I see this information in Table 3; however, the difference in the AIC is only 1.2(!!) and it is unclear how variables were retained (e.g., "decision to reject sex" has a p-value in the full model that is far superior to the p-value for PrEP awareness); moreover, it is no longer best practice to simply pare down models based on simple p-value analyses. For example, I would expect strong collinearity between "PrEP knowledge" and "PrEP awareness" and even potentially "decision to have safe sex" - more information must be provided about how these model decisions were made.

5. You interpret ORs from the full model, which in my opinion has FAR too many variables, many of which are likely collinear. This does not seem to be an appropriate use of modeling.

6. What is the value to the field of noting (in the second paragraph of the discussion) that before the Netherlands implemented PrEP in a systematic way, nationwide, there was low uptake of PrEP? Is this not obvious?

7. You do not use the Health Belief Model as a framework at all in this paper until the Discussion, where you say that being aware of PrEP and having intention to take PrEP may facilitate the uptake of PrEP in the Netherlands, which may be explained by the Health Belief Model. Frankly such a finding is explained by simple logic (one is very unlikely to initiate PrEP if they don't know about it and/or don't plan to take it), and does not require modeling to illuminate. However, if you plan to reference the Health Belief Model you have a responsibility to fundamentally use it and explicitly draw the connections you are making to the framework. 

Minor comments:

1. The EMIS survey should be spelled out/explained in the abstract.

Round 2

Reviewer 3 Report

Dear authors, I was quite impressed with the responsiveness to my initial review, and improvements to the manuscript. I believe it now reads much more clearly, with coherent presentation of background, methods, and findings. Nice work.

I remain concerned only with the manual stepwise backwards selection procedure, which is obviously too late to change at this stage. I didn't provide much detail to my objection before given the other fundamental problems with the paper, but as the paper is much improved I do feel it's worth raising again. While still very frequently done, using p-values to select final models in the way you've done here is problematic because p-values represent the probability of seeing a test statistic at least as extreme as the one you have, when H0 is true (in this case, the p-value should be uniformly distributed). But after stepwise selection, the p-values of the retained variables don't have that property, because you've chosen the variables that have small p-values, which means the variables left in the model are unusually deflated. The p-value also conflates both the size of the coefficient with the quality of its measurement, which is an inappropriate measure by which to select the relevance of various predictors to an outcome. More information on this can be found here: https://www.stat.cmu.edu/~cshalizi/mreg/15/lectures/26/lecture-26.pdf.

Anyway, as I said, this won't reasonably be changed now, and given how common it still is as a practice, I don't think it's reason to hold up publication of the article. However, I recommend changing one sentence: "We retained all variables with p<0.05 to ensure adjustment for potential confounding variables." It is simply erroneous to suggest that since you used a manual stepwise backwards selection procedure with a cutoff of p<0.05 you are ensuring adjustment for potential confounding variables. Uncontrolled confounding is almost always a limitation, and confounding is best minimized through model selection based on subject matter expertise (e.g. a thoughtful DAG) and not statistical testing of this nature. Please revise. 

Author Response

Dear reviewer, thanks for your comments and fair judgment. We are happy to hear that you deem the revision to be improved. We have adjusted the sentence accordingly, now only stating the cut-off.